# Zinc(II)-Sterol Hydrazone Complex as a Potent Anti-*Leishmania* Agent: Synthesis, Characterization, and Insight into Its Mechanism of Antiparasitic Action

**DOI:** 10.3390/pharmaceutics15041113

**Published:** 2023-03-31

**Authors:** Gonzalo Visbal, Rodrigo M. S. Justo, Gabrielle dos Santos da Silva e Miranda, Sara Teixeira de Macedo Silva, Wanderley de Souza, Juliany Cola Fernandes Rodrigues, Maribel Navarro

**Affiliations:** 1Laboratório de Ácidos Nucleicos (Laban), Coordenação Geral de Biologia (Cobio), Diretoria de Metrologia Científica e Industrial, DIMCI, Instituto Nacional de Metrologia, Qualidade e Tecnologia, INMETRO, Rio de Janeiro 25250-020, Brazil; 2Laboratório de Químicas Bioinorgânica e Catalise (LaQBIC), Departamento de Química, Instituto de Ciências Exatas, Universidade Federal de Juiz de Fora Juiz De Fora, Juiz de Fora 36036-900, Brazil; rodrigo.manoel1992@gmail.com; 3Núcleo Multidisciplinar de Pesquisa em Biologia (NUMPEX-Bio), Campus UFRJ-Duque de Caxias Prof. Geraldo Cidade, Universidade Federal do Rio de Janeiro, Rio de Janeiro 21941-901, Brazil; gabrielle.biotec@gmail.com (G.d.S.d.S.e.M.); juliany.rodrigues@xerem.ufrj.br (J.C.F.R.); 4Centro Nacional de Biologia Estrutural e Bioimagem, CENABIO, Universidade Federal do Rio de Janeiro, Rio de Janeiro 21941-901, Brazil; sara.teixeiracp2@gmail.com; 5Laboratório de Ultraestrutura Celular Hertha Meyer, Instituto de Biofísica Carlos Chagas Filho, Universidade Federal do Rio de Janeiro, Rio de Janeiro 21941-901, Brazil; wsouza@biof.ufrj.br; 6Centro Multiusuário para Análise de Fenômenos Biológicos, Escola Superior de Saúde, Universidade do Estado do Amazonas, Manaus 69850-000, Brazil

**Keywords:** zinc, sterol hydrazone, *Leishmania*, sterol methyltransferase, parasites, metal-drug

## Abstract

Searching for new alternatives for treating leishmaniasis, we present the synthesis, characterization, and biological evaluation against *Leishmania amazonensis* of the new ZnCl_2_(**H3**)_2_ complex. **H3** is 22-hydrazone-imidazoline-2-yl-chol-5-ene-3β-ol, a well-known bioactive molecule functioning as a sterol Δ^24^-sterol methyl transferase (24-SMT) inhibitor. The ZnCl_2_(**H3**)_2_ complex was characterized by infrared, UV-vis, molar conductance measurements, elemental analysis, mass spectrometry, and NMR experiments. The biological results showed that the free ligand **H3** and ZnCl_2_(**H3**)_2_ significantly inhibited the growth of promastigotes and intracellular amastigotes. The IC_50_ values found for **H3** and ZnCl_2_(**H3**)_2_ were 5.2 µM and 2.5 µM for promastigotes, and 543 nM and 32 nM for intracellular amastigotes, respectively. Thus, the ZnCl_2_(**H3**)_2_ complex proved to be seventeen times more potent than the free ligand **H3** against the intracellular amastigote, the clinically relevant stage. Furthermore, cytotoxicity assays and determination of selectivity index (SI) revealed that ZnCl_2_(**H3**)_2_ (CC_50_ = 5 μΜ, SI = 156) is more selective than **H3** (CC_50_ = 10 μΜ, SI = 20). Furthermore, as **H3** is a specific inhibitor of the 24-SMT, free sterol analysis was performed. The results showed that **H3** was not only able to induce depletion of endogenous parasite sterols (episterol and 5-dehydroepisterol) and their replacement by 24-desalkyl sterols (cholesta-5,7,24-trien-3β-ol and cholesta-7,24-dien-3β-ol) but also its zinc derivative resulting in a loss of cell viability. Using electron microscopy, studies on the fine ultrastructure of the parasites showed significant differences between the control cells and parasites treated with **H3** and ZnCl_2_(**H3**)_2_. The inhibitors induced membrane wrinkle, mitochondrial injury, and abnormal chromatin condensation changes that are more intense in the cells treated with ZnCl_2_(**H3**)_2_.

## 1. Introduction

*Leishmania* spp. are protozoan parasites of the Trypanosomatidae family that exhibit two developmental stages during their life cycle, alternating between the insect vector and the vertebrate host [1]. Leishmaniasis is a complex of severe diseases that can be mutilated and, in the case of visceral leishmaniasis, if not diagnosed rapidly, could lead to death [2]. Furthermore, leishmaniasis is one of the most important neglected tropical diseases, where around 1 billion people worldwide are at risk for infection [3]. The available treatments for leishmaniasis are limited by drawbacks, such as chemoresistance, cardiotoxicity, nephrotoxicity, hepatotoxicity, diarrhea, headaches, and low selectivity, among other side effects [4]. Therefore, there is an urge to seek new chemotherapeutics against leishmaniasis and other neglected diseases. Over the years, metal-based drugs against neglected diseases have been discussed as alternative ways to overcome the difficulties presented by approved antiparasitic agents. The study of later transition metal-based drugs as chemotherapeutics is an exciting research field in chemistry, biology, and medicine due to the ability to develop multitarget antiparasitic agents [5,6].

In this fashion, sterol biosynthesis, an ancient metabolic pathway that plays a key role in the diversification and evolution of eukaryotic organisms, became a targeting specific [6]. Furthermore, this metabolic pathway has shown essential enzymes for the trypanosomatids’ survival, leading to the rational development of new drugs for treating leishmaniasis and trypanosomiases. The most prominent examples on the list comprise HMG-CoA reductase, which in humans serves as a clinical target for statins [7], farnesyl diphosphate synthase, the target for bisphosphonates [8], squalene synthase, which is inhibited by quinuclidine derivatives [9,10], and sterol 14α-demethylase as a clinical target for azoles [11] and sterol Δ^24^-sterol methyl transferase (24-SMT) inhibited by azasterols (this enzyme is unique for human pathogens and not present in humans) [6].

The use of azasterol derivatives as an alternative for treating leishmaniasis came up with the inhibition activity towards the enzyme 24-SMT, which plays an essential role in parasite sterol biosynthesis (PSB) [6,12]. Episterol and 5-dehydroepisterol are the main sterols found in *Leishmania* parasite cells, controlling cell permeability and fluidity [5,6]. The need to develop 24-SMT inhibitors is held on the fact that in vertebrates, 24-SMT is not an enzyme found in cholesterol biosynthesis; therefore, a 24-SMT inhibitor would selectively inhibit PSB without damaging mammalian host cells [6,13,14]. Several studies developed by Nes, W.D research groups revealed that the presence of a heteroatom in steroid-like structures proved effective in inhibiting 24-SMT, especially in which the heteroatom, nitrogen or sulfur, is found at carbons C-24 and C-25 [12]. The presence of a heteroatom in the sterol structure would contribute to the drug efficiency due to the possibility of mimicking intermediates that can bind strongly to the 24-SMT enzyme [12,15]. Among the steroid-like structures, azasterol derivatives (nitrogen-containing sterol compounds) were already reported as inhibitors of 24-SMT in fungi and protozoans by avoiding the methylation of zymosterol (in *Leishmania* spp.) or lanosterol (in fungi) [6,16]. The search for new classes of azasterol-derivatives is still ongoing, and the association of sterol hydrazones with metals is exciting when considering the development of new antiparasitic metallodrugs [5].

Metal-azasterol complexes carry the concept of metal–drug synergism. This strategy helps enhance the new drug’s activity due to the binding capability and reactivity of the transition metal, therefore, granting notable benefits: (i) longer organism-time residence; (ii) efficiently targets reach; and (iii) decrease in toxicity [17]. Previously reported works showed the efficacy of bioactive azole compounds (clotrimazole, ketoconazole, and itraconazole) coordinated with metals toward the treatment of fungal infections [18,19] and protozoa illnesses [5,20]. Regarding using of metal-azasterol complexes as antiparasitic agents, platinum was employed in association with azasterol derivatives to treat *Leishmania mexicana* proliferation, proving that the metal–bioactive drug association was able to increase anti-*Leishmania* activity over the free ligand [5]. Additionally, azasterol derivatives have been coordinated with copper and gold, and their antifungal activity was evaluated against *Sporothrix schenkii* and *Sporothrix brasiliensis* yeasts [21].

The use of metals, such as platinum, gold, titanium, and ruthenium, has been extensively explored by several groups for the development of metallodrug agents to treat cancer, HIV, and parasitic disease [5,20,22,23,24,25,26,27]. However, there have only been a few studies of zinc as a possible chemotherapeutic agent [5,18,27].

Zinc is considered an essential metal for the life and health of humans [28,29] being applied also as a pharmaceutical agent in chemotherapy against degenerative and genetic diseases (cancer and diabetes) [30]. In addition, the incorporation of bioactive ligands to zinc to obtain drugs with biological activity towards parasitic infections, such as Leishmaniasis, Chagas disease, and *Sporothrix* sp., have been reported in the literature [5,18,20,27]. A recent work developed by our group, which shows the use of trace metals in association with azasterol derivatives, described the synthesis of copper and zinc-azasterol complexes and their antifungal activity against *Sporothrix brasiliensis* [31]. However, the use of zinc–azasterol complexes as anti-*Leishmania* agents has yet to be reported; therefore, our interest is to provide the synthesis and characterization of a new ZnCl_2_(**H3**)_2_, or also called here Zn**H3** complex, as well as its biological activity against *Leishmania amazonensis* [32].

## 2. Experimental Section

### 2.1. Materials and Methods

Organic solvents used in the reactions were purchased from Supelco^®^ and Emsure^®^ and the zinc chloride from Vetec. The new zinc complex was characterized using the following spectroscopic and analytical techniques: IR spectra were obtained with a Brucker alpha ATR-IR Spectrometer in the range of 4000–400 cm^−1^. UV-Vis spectra were obtained with a Shimadzu UV-1800 spectrometer in DMSO in the range of 190–1100 nm. Elemental analyses were obtained using a micro analyzer (CHNS/O) FLASH 2000 model from Thermo Scientific. Conductivity values were determined with a MS Tecnopon NI-CVM instrument. NMR experiments were performed at 298 K on a Bruker Avance III HD 500 spectrometer, 9.4 T in DMSO-d_6_ solutions.

### 2.2. Synthesis of ZnCl_2_(**H3**)_2_

To a solution of ZnCl_2_ (16.36 mg, 0.12 mmol) in methanol (12 mL), 22-hydrazone-imidazolin-2-yl-chol-5-ene-3β-ol (**H3**) (100 mg, 0.24 mmol) in methanol (24 mL) was added dropwise, and the reaction mixture remained colorless. The system was heated under reflux for 24 h. After 24 h, a white solid precipitate was formed, and the reaction was cooled off to room temperature. The white precipitate was filtered off, washed twice with distilled water and diethyl ether, and dried under a vacuum. Yield 44% (51.3 mg). Elemental analysis (%) calculated for Zn(C_24_H_38_N_4_O)_2_Cl_2_
^.^ H_2_O ^.^ CH_3_OH: C 60,24; H 8.57, N: 11.08, found C 60.24, H 7.86, N 10.68. ESI-MS (*m*/*z*); [M] = 958.62. Molar conductivity in DMSO, Λ_M_ = 8.52 ohm^−1^cm^2^mol^−1^. IR (cm^−1^): ν(NH, OH) 3454, 3318; ν(C_sp_3-H) 2935, 2899; ν(C=N) 1667. λ_max_ nm in MeOH (ɛ, M^−1^cm^−1^): 226 (π→π*, 17,433). ^1^H-NMR (DMSO-*d6*) [δ ppm, (integral, assignation)]: 7.37 (s, 1H, H22); 5.27 (s, 1H, H6); 3.25 (m, H24, H25); 1.04 (d, *J* = 6.60 Hz, 3H, H21); 0.95 (s, 3H, H19); 0.68 (s, 3H, H18). ^13^C-NMR (DMSO-*d6*) [δ ppm (assignation)]: 159.79 (C-23); 156.86 (C-22); 120.83 (C-6); 42.68 e 42.60 (C-24 e C-25); 19.62 (C-19); 12.43 (C-18).

### 2.3. Biological Assays

Parasites. The *Leishmania amazonensis* WHOM/BR/75/JOSEFA strain was used for the biological assays. This strain was isolated from a patient with diffuse cutaneous leishmaniasis by Dr. Cesar A. Cuba-Cuba (Brasilia University, Brazil, 1975) and kindly provided by the *Leishmania* Collection of the Instituto Oswaldo Cruz (Code IOCL 0071-FIOCRUZ). The infectivity of the parasites was maintained by inoculating metacyclic promastigotes at the base of the tail of BALB/c mice. Intracellular amastigotes obtained from the lesion were differentiated into promastigotes, which were maintained in Warren’s medium (brain heart infusion plus hemin and folic acid) supplemented with 10% fetal bovine serum at 25 °C. Infective metacyclic promastigotes from cultures in the stationary phase were used to infect murine macrophages to obtain intracellular amastigotes.

Drugs. For the biological assays, **H3** and Zn-**H3** were dissolved in dimethyl sulfoxide (DMSO), and the maximum DMSO concentration in the cultures did not exceed 0.5%, a concentration that did not interfere with the *Leishmania* growth. The compound’s solution was stored at −20 °C.

Antiproliferative effects on *L. amazonensis* promastigotes and intracellular amastigotes. *L. amazonensis* promastigotes growth curves were initiated with 1 × 10^6^ cells/mL in Warren’s medium supplemented with 10% fetal bovine serum. After 24 h of growth, different concentrations of **H3** [1; 3; 4; 5; 6; 7; 8 μM] and Zn-**H3** [0.5; 1; 2; 3 μM] were added, and cell density was determined daily by counting in a Neubauer chamber using contrast phase light microscopy for 96 h. For intracellular amastigotes, monocytes were obtained through peritoneal lavage of BALB/c mice with Hank’s solution (pH 7.2) and placed to adhere in 24-well plates for 30 min at 37 °C with 5% CO_2_. After that, the wells were washed to remove unadhered cells, and the plates were maintained for 24 h. Then, metacyclic promastigotes were added to interact with macrophages for 2 h at a ratio of 10:1. The parasites not internalized were removed after being washed with culture medium, and the infected culture was maintained in RPMI supplemented with 10% fetal bovine serum for 24 h. Different concentrations of **H3** [1; 3; 5; 8 μM] and Zn-**H3** [0.5; 1; 3; 5; 8 μM] were added, and the fresh medium (RPMI, pH 7.2) with the inhibitors was changed every 24 h. After 48 h of treatment, the infected cultures were fixed with 4% freshly prepared formaldehyde in phosphate buffer saline (PBS, pH 7.2) and stained with Giemsa for 15 min. The percentage of infected cells was determined by light microscopy. Association indexes (mean number of parasites internalized per cell, multiplied by the percentage of infected macrophages, and divided by the total number of macrophages) were determined and used as a parameter to calculate the percentage of infection in each condition used in this study. The concentration to inhibit 50% of the infection (IC_50_) was calculated using the GraphPad Prism 6 software. The results are expressed as the means of three independent experiments.

Cell viability of promastigotes and peritoneal macrophages. The cell viability for promastigotes and peritoneal macrophages treated with **H3** and Zn-**H3** was determined using the CellTiter 96^®^ Aqueous MTS Assay(Promega, São Paulo, Brasil) [33]. The promastigotes cultures were initiated with a cell density of 1 × 10^6^ cells/mL in Warren’s medium supplemented with 10% fetal bovine serum. After 24 h of growth, different concentrations of **H3** and Zn-**H3** were added. After 48 h of treatment, each group (treated and untreated) was transferred in triplicate to the 96-well plate. Peritoneal macrophages were obtained from BALB/C mice, as described in the previous section, and cultured in triplicate in 96-well plates. Macrophages were incubated with 5, 10, 15, 20, 30, and 40 µM **H3** for 48 h. MTS/PMS assay reaction was quantified by optical density measurement at 490 nm in a microplate reader SpectraMax M2/M2^e^ (Molecular Devices, San Jose, CA, USA). Data were plotted and subjected to statistical analysis using GraphPad Prism 6 software.

Measurements of reactive oxygen species. The production of reactive oxygen species was analyzed using the fluorescent probe CM-H_2_DCFDA. For this, 1 mL of control, **H3**, and Zn-**H3**-treated promastigotes for 48 h were collected and washed with PBS-glucose, pH 7.2. Next, promastigotes were resuspended in a solution containing 10 µM CM-H_2_DCFDA in the same buffer and incubated for 20 min at 25 °C. After that, cells were washed and resuspended in 500 µL PBS-glucose, pH 7.2. Finally, as a positive control, control cells were incubated with 0.5% hydrogen peroxide in PBS-glucose, pH 7.2, for 20 min before labeling with CM-H2DCFDA. After this, samples were analyzed by flow cytometry in an Accuri C6 cytometer, San Jose, CA, USA; 50.000 events were evaluated, then the data were plotted and subjected to statistical analysis using GraphPad Prism 6 software.

Estimation of mitochondrial transmembrane electric potential (ΔΨ*m*). The mitochondrial membrane potential (ΔΨ*m*) was analyzed with JC-1 fluorochrome. For this, control and treated promastigotes were washed with PBS, pH 7.2, and maintained in the reaction medium containing 125 mM sucrose, 65 mM KCl, 10 mM HEPES/K^+^, pH 7.2, 2 mM propidium iodide (Pi), 1 mM MgCl_2_ and 500 µM EGTA. Thus, 1 × 10^7^ cells/mL were incubated with 10 µg/mL JC-1 for 40 min at 25 °C with readings made every minute using a SpectraMax M2/M2^e^ microplate reader (Molecular Devices, San Jose, CA, USA). After 24 min, 2 µM FCCP was added to abolish the ΔΨ*m* sustained at the inner mitochondrial membrane by the respiratory chain. Relative ΔΨ*m* was obtained by calculating the ratio between the reading at 590 nm and 530 nm (590:530 ratio). Results obtained from each triplicate were analyzed and statistically analyzed using the GraphPad Prism 6 software; the data shown in the figures represent these experiments.

Electron microscopy. Scanning and transmission electron microscopy were carried out to study the morphological and ultrastructural alterations induced by treating *L. amazonensis* promastigotes with **H3** and Zn-**H3**. Control and treated promastigotes were firstly fixed with 2.5% glutaraldehyde type I in 0.1 M cacodylate buffer pH 7.2 and washed three times with 0.1 cacodylate buffer. For scanning electron microscopy, parasites were adhered to coverslips previously covered with poly-L-lysine, postfixed in a solution containing 1% OsO_4_, 1.25% potassium ferrocyanide, and 5 mM CaCl_2_ in 0.1 M cacodylate buffer pH 7.2 for 30 min. After that, samples were dehydrated in increased ethanol concentrations (30, 50, 70, 90, and 100%), critical point-dried in CO_2_, sputtered with a thin gold layer, and observed under a FEI Quanta 2050 scanning electron microscope. For transmission electron microscopy, parasites were postfixed in the same solution, washed three times in 0.1 M cacodylate buffer pH 7.2, dehydrated in increased acetone concentrations (30, 50, 70, 90, and 100%), and embedded in epoxy resin. Ultrathin sections were obtained, contrasted with uranyl acetate and lead citrate, and observed under a Zeiss 900 electron microscope (Jena, Germany).

Ethics approval. The experiments using BALB/c mice to isolate macrophages and to maintain *Leishmania* parasites were approved by the Ethics Committee for Animal Experimentation of the Health Sciences Centre, Federal University of Rio de Janeiro (Protocols n. IBCCF 096/097/106), according to the Brazilian Federal Law (11.794/2008, Decreto no 6.899/2009). Furthermore, all animals received humane care in compliance with the “Principles of Laboratory Animal Care” formulated by the National Society for Medical Research and the “Guide for the Care and Use of Laboratory Animals” prepared by the National Academy of Sciences, USA. The experiment involving animals follows the recommendations described in the ARRIVE guidelines.

Statistical analysis. The means and standard deviations were calculated for experiments performed in triplicate (except for electron microscopy investigation). A One-way ANOVA with Dunnett’s post hoc test was used to analyze the antiproliferative effect, the MTS/PMS assays, ROS production, and the mitochondrial transmembrane electric potential estimation. Different *p* values were obtained for statistical analyses, which are mentioned in the legend of the figures.

Extraction and separation of neutral lipids. These procedures have been described previously [34]. Briefly, *L. amazonensis* was cultured as described above in the presence of increasing concentrations (2 to 7 µM) of **H3** and ZnCl_2_(**H3**)_2_. Lipids were extracted with chloroform-methanol (2:1, *v*/*v*). The extract was dried and suspended in a minimum volume of chloroform. The chloroform suspension was applied to a silicic acid column (1.5 by 4 cm) and washed with 4 column volumes of chloroform to separate neutral lipids from other lipid fractions.

Free sterols analysis. For analysis and structural assignment, neutral lipids were separated in a high-resolution capillary column (Ultra-2, 25 m × 0.20 mm i.d., with 5% phenyl-methylsiloxane and a film thickness of 0.33 μm) in an Agilent Technologies 7890A gas chromatograph equipped with a 5975C inert XL MSD mass selective detector (Agilent Technologies, Inc. Santa Clara, CA, USA). Lipids were dissolved in ethyl acetate and injected into the column at an initial temperature of 50 °C (1 min), followed by a temperature increase to 270 °C at a rate of 20 °C/min and a further increase to 290 °C at a rate of 1 °C/min. The carrier gas (He) flow was kept constant at 1 mL/min. The injector temperature was 250 °C, and the detector was kept at 280 °C. The total run time was 53 min. Mass spectra were obtained by electron ionization (EI) at 70° eV. The assignment of structures was based on relative chromatographic behavior and the characteristic fragmentation patterns observed in MS and by comparison of the mass spectra with those available in the NIST library.

## 3. Results and Discussions

### 3.1. Chemical Section

Azasterol **H3** was prepared according to a previously reported procedure [15]. The metal complex ZnCl_2_(**H3**)_2_ was synthesized through mild conditions reaction (Figure 1). The free ligand **H3** was dissolved in methanol and added dropwise to a methanolic solution of ZnCl_2_ at a molar ratio of 2:1 ligand:metal. ZnCl_2_(**H3**)_2_ was obtained as an off-white solid and characterized through spectroscopic and analytic techniques.

Mass spectrometry (ESI-MS) of the ZnCl_2_(**H3**)_2_ complex displayed a molecular ion peak [M]^+^ observed at *m/z* 958.62, and the peak corresponding to the free ligand, **H3**, was observed at *m/z* 413,30 [**H3**+H]^+^. Figure 1 shows fragments of the ESI-MS spectrum of the ZnCl_2_(**H3**)_2_ complex.

The molecular formula for the complex was attributed according to molecular conductivity measurements and elemental analysis experimentation. ZnCl_2_(**H3**)_2_ was a neutral complex; therefore, chloride ions were assumed as ligands and not counter ions [35]. Moreover, elemental analysis data also showed a proportion of two molecules of the ligand **H3** for one molecule fragment of ZnCl_2_. Thus, the data obtained agrees with the molecular formula proposed.

UV-Vis spectra of the free ligand **H3** and Zn–**H3** complex were acquired using methanol as a solvent. Regarding the azasterol **H3**, an intense band at 227.2 nm, characteristic of the steroidal fragment and assigned to π-π* transitions, was observed [36]. The electronic spectrum of ZnCl_2_(**H3**)_2_ also exhibited an intense band at 226 nm with a molar absorptivity (ε) value of 17,433 M^−1^cm^−1^, characteristic of the ligand. A higher magnitude of ε indicates the presence of ligand-to-metal charge transference (LCMT) absorptions, characteristic of metal coordinated to nitrogenated ligands. Absorption spectra of the free ligand **H3** and Zn–**H3** complex can be seen in the supporting information (Appendix A).

The IR spectrum of the Zn–**H3** complex showed characteristic bands associated with the free ligand **H3**. Broad bands corresponding to the ν(NH, OH) stretching were observed at 3454 and 3318 cm^−1^, while bands at 2935 cm^−1^ and 2899 cm^−1^ were assigned to the ν(C_sp3_-H) stretching. The characteristic band of C=N stretching, present in the side chain of the azasterol, was observed at 1667 cm^−1^. The C=N absorption stretching was shifted to a higher frequency region (20 cm^−1^) compared to the free sterol hydrazone, which possibly indicates the coordination of the ligand **H3** to the metal ion Zn(II) through the C=N group. Additionally, a small intensity band was observed at 1056 cm^−1^, corresponding to the C-O stretching found in the steroidal fragment. The IR spectrum of the Zn–**H3** complex can be found in the supporting information (Appendix A).

^1^H and ^13^C NMR characterization of complex ZnCl_2_(**H3**)_2_ was carried out using DMSO-*d6* as a solvent, and chemical shift variation (Δδ) of each signal concerning those of the free ligand was taken as a parameter to deduce the mode of binding of **H3** to the zinc(II) ion. ^1^H NMR spectrum of the Zn–**H3** complex exhibited all signals characteristic of the azasterol in the supporting information (Appendix A). Singlet peaks at 0.68 and 0.95 ppm were assigned to the methyl hydrogens **H_18_** and **H_19_** from the steroidal fragment. In contrast, a doublet at 1.04 ppm (*J* = 6.60 Hz) was ascribed to the methyl hydrogen **H_21_**, belonging to the side chain of the ligand coordinated to Zn(II). Signals assigned to the hydrogens from the alicyclic moiety of the azasterol were found in the range of 1.0–2.4 ppm. **H_24_** and **H_25_** were assigned as multiplet centered at 3.25 ppm. Regarding the steroidal fragment, olefinic hydrogen **H_6_** could be observed as a broad singlet at 5.27 ppm. The iminic hydrogen **H_22_** from the C=N bond could be seen as a singlet at 7.37 ppm. The coordination of the azasterol **H3** to the metal ion Zn(II) promoted changes in the behavior of the signals observed in the ^1^H NMR spectrum. Chemical shift variations regarding the steroidal moiety of the Zn–**H3** complex were insignificant; however, the iminic hydrogen **H_22_** displayed a downfield chemical shift of 0.23 ppm. This slight change also indicates possible coordination of the ligand **H3** to the metal ion through the C=N bond.

^13^C NMR of the complex ZnCl_2_(**H3**)_2_ also exhibited characteristic peaks of the ligand used, and some changes must be discussed. Methyl carbons **C_18_**, **C_21_**, and **C_19_** were observed as high-intensity peaks at 12.43, 17.92, and 19.62 ppm, respectively. Peaks found at 70.47, 120.83, and 141.78 ppm were assigned to the carbons **C_3_**, **C_6_**, and **C_5_**, respectively. Low-intensity peaks centered at 156.86 and 159.79 ppm corresponded to the iminic carbons **C_22_** and **C_23_**, respectively. Iminic carbons **C_22_** and **C_23_** exhibited more significative chemical shift variations when the ligand **H3** was coordinated to the metal ion Zn(II), in which **C_22_** shifted 2.56 ppm downfield (higher frequency) and **C_23_** shifted 6.25 ppm upfield (lower frequency). Hence, based on previously reported studies and the acquired ^1^H and ^13^C NMR data, the coordination site proposed for the azasterol **H3** occurs through the nitrogen atom from the imine carbon **C_23_**. The ^13^C NMR spectra comparison can be seen in Figure 2.

The stability of ZnCl_2_(**H3**)_2_ complex was examined by ^1^H NMR for six days using DMSO-d6 as a solvent. The spectra of this study are shown in Appendix A. ^1^H NMR spectra showed no changes in hydrogen signals for the compound ZnCl2(**H3**)2, which means all spectra appeared to be the same despite the time analysis. These results highlight the stability of the synthesized Zn–**H3** complex, which suggests that the complex does not suffer any dissociation or ligand exchange in DMSO.

### 3.2. Biological Activity

#### 3.2.1. Antiproliferative, Ultrastructural, and Physiological Studies on *L. amazonensis* Promastigotes and Intracellular Amastigotes

Biological assays were performed on *Leishmania amazonensis* promastigotes and intracellular amastigotes. The promastigotes were susceptible to the treatments, presenting IC_50_ values of 5.17 µM and 2.47 µM for **H3** and the Zn–**H3** complex after 48 h (Figure 3A,B). In addition, the MTS/PMS assay was used to analyze the promastigotes’ cell viability, measuring the mitochondrial dehydrogenase activity in metabolically active cells. Figure 3C shows that, close to the IC_50_ values, there is an abrupt reduction in the number of parasites and cell viability, which is even more evident in the concentration of 8 µM **H3** and 5 µM Zn-**H3** (Figure 3C). Furthermore, it is possible to infer that one of the cellular targets of the treatments is the mitochondria, directly affecting the activity of mitochondrial dehydrogenases.

In the intracellular amastigotes, the results were even more promising than the promastigotes (Figure 3), increasing 9 and 75 times the efficiency of **H3** and Zn-**H3**, respectively (Figure 4). Furthermore, after 48 h of treatment, the IC_50_ values of 543.42 nM and 32.8 nM were observed for **H3** and Zn-**H3**, respectively (Figure 4), revealing that the presence of the metal resulted in the improvement of the compound, which in this case was seventeen times. Thus, the antiproliferative analysis revealed that the cation Zn(II) enhances the biological activity of the azasterol derivative **H3** in both developmental stages (Figure 3 and Figure 4), highlighting the concept of metal–drug synergism. It is important to mention here that our group already showed that there was no significant effect against promastigotes for zinc salts used for the synthesis process [5].

The cytotoxicity activity of **H3** was evaluated by studying its effect on the murine macrophages (Figure 5). The results revealed that the compound **H3** reduced 50% of the cell viability at around 5 μM. Thus, resulting in a CC_50_ value that was much superior to the IC_50_ value obtained for intracellular amastigotes, highlighting the low cytotoxic activity of the **H3**.

Analyzing the antiproliferative effects and comparing them with other azasterols [11,37], the results revealed that Zn-**H3** was much more efficient in inhibiting the parasite growth, mainly the intracellular amastigotes. For example, the IC_50_ value (32.8 nM) for the metallodrug was lower than the value found for 22,26-azasterol (around 100 nM), one of the first azasterols evaluated against *L. amazonensis* [37].

After the prominent antiproliferative effects induced by **H3** and Zn-**H3**, we decided to use electron microscopy to identify organelles and structures affected by this class of drug. The analysis of the promastigote morphology by scanning electron microscopy (Figure 6A,C,D) showed significant differences between parasites treated with **H3** and Zn-**H3** (Figure 6C,E). Promastigotes treated with **H3** presented a thinning cell body in its posterior region (Figure 6C, arrowhead). At the same time, Zn-**H3** produced rounded parasites (Figure 6E, thick arrows) that lost the fusiform shape characteristic of the promastigotes. In addition, a significant wrinkling was also observed, more intensely, in the parasites treated with Zn-**H3** (Figure 6E, thin arrow). Thus, a substantial difference in the cell morphology can be observed when comparing the treated promastigotes with the control cells (Figure 6A).

On the other hand, the images obtained by transmission electron microscopy (Figure 6B,D,F) also revealed basic information about the cellular ultrastructure. Figure 6B shows a control promastigote presenting its characteristic fusiform cell body with well-preserved intracellular structures and organelles, such as the nucleus, mitochondrion, and lipid bodies. After treatment with **H3** and Zn-**H3**, several ultrastructural alterations were observed, such as (1) disorganization of kDNA (Figure 6D); (2) mitochondrial swelling with loss of the matrix content (Figure 6D,F); and (3) presence of myelin-like figures and large autophagic vacuoles (Figure 6F). These alterations are typically found in protozoan parasites treated with 24-SMT inhibitors [6,37]. Due to the ultrastructural alterations observed after treatment with **H3** and Zn-**H3**, two analyses of the mitochondrial function and cell physiology were carried out: mitochondrial membrane potential (ΔΨ*m*) and reactive oxygen species (ROS) production. Fluorescence analysis using the JC-1 revealed a strong reduction of the mitochondrial membrane potential, especially in treatments with the metallodrug Zn-**H3** (Figure 7), an alteration that corroborates with the mitochondrial swelling and loss of the mitochondrial matrix. Several studies in *Leishmania* and other trypanosomatids have demonstrated that different molecules impact mitochondrial function, including sterol biosynthesis inhibitors [38,39]. Regarding the sterol composition, the trypanosomatid’s mitochondrion has a unique lipid content compared with mammalian cells [38,39]. These effects on the mitochondrion corroborate with results obtained by the MTS/PMS assay, indicating that this organelle is a cellular target of **H3** and Zn-**H3**.

In addition, ROS production was evaluated using the H_2_DCFDA marker. In all treatments, an increase in ROS production was observed, mainly at the concentration of 5 μM Zn-**H3** (Figure 8). This data corroborates with the cellular stress observed by the studies with electron microscopy and about mitochondrial function, indicating that the mitochondrion and endoplasmic reticulum are the main organelle targets. In addition, alterations in these organelles can also be related to inhibiting the enzymes involved in the ergosterol biosynthesis pathway since some enzymes are in them [40,41]. Therefore, mitochondrion and endoplasmic reticulum (ER) are also involved in autophagic processes, which after the treatments, could be induced, especially to remove sterol intermediate accumulated. Thus, ER is one of the critical organelles that is directly and indirectly affected during the treatment of promastigotes with these azasterol derivatives.

Thus, considering the impacts on the ultrastructure of the treated parasites and the reduction of cell viability, together with the production of reactive oxygen species and damage to the mitochondrial membrane potential, these alterations reflect the possible mechanisms of action of the inhibitors studied here.

#### 3.2.2. Free Sterol Analyses of *L. amazonensis* Promastigotes Control and Treated with **H3** and ZnCl_2_(**H3**)_2_

The free sterol composition of *Leishmania amazonensis* promastigotes measured by high-resolution capillary gas chromatography coupled to mass spectrometry (GC–MS) is presented in detail (Table 1 and Table 2, and Figure 9). In control (untreated) promastigotes, the predominant sterols were ergosta-5,7,24(24′)-trien-3β-ol (5-dehydroepisterol) and ergosta-7,24(24′)-dien-3β-ol (episterol), both synthesized de novo representing 80% of the total sterols. While other minority sterols were identified, such as zymosterol, cholesta-5,7,24-trien-3β-ol, cholesta-7,24-trien-3β-ol, ergosta-5,7,9(11)- 24(24′)-tetraen-3β-ol, ergosta-7,22,24(24′)-trien-3β-ol, ergosta-5,7,22,24(24′)-teraen-3β-ol, ergosta-5,8,22-trien-3β-ol, ergosta-5,24(24′)-dien-3β-ol, 14α-metil-ergosta-8,24(24′)-dien-3β-ol, ergosta-5,7,9(11)-24(24′)-tetraen-3β-ol, lanosterol, estigmasta-5,7,22-trien-3β-ol, and estigmasta-7,22-trien-3β-ol, they represented about 13% of the total. Cholesterol, incorporated into the parasite by endocytosis from the growth medium, accounted for 7%.

Incubation of promastigotes with increasing concentrations of **H3** (3–7 µM) or ZnCl_2_(**H3**)_2_ (3–7 µM) showed a concentration-dependent growth arrest, coinciding with a substantial reduction in the proportion of 24-alkylated sterols (namely, episterol and 5-dehydroepisterol) from approximately 80% to 5% and a corresponding increase in the proportion of nonalkylated sterols, such as cholesta-5,7,24-trien-3β-ol and cholesta-7,24-dien-3β-ol, in 84%, suggestive of 24-SMT inhibition (Table 1 and Table 2, and Figure 9). These results also coincide with those obtained by Rodrigues et al. and Lorente et al. when they studied the biological activity of azasterols on *L. amazonensis* promastigotes [6,37]. It should be noted that ZnCl_2_(**H3**)_2_, at a low concentration of 2 μΜ, caused an accumulation of cholesta-type sterols in 81%. This outcome suggests that ZnCl_2_(**H3**)_2_ also significantly inhibits the enzyme 24-SMT leading to a stronger antiproliferative effect by disrupting 5-dehydroepisterol and episterol homeostasis (Figure 9). The numbers found in the columns for control and treated parasites represent the percentage of the total sterols.

## 4. Conclusions

A new zinc–azasterol complex, ZnCl_2_(**H3**)_2_, was prepared and fully characterized using analytical and spectroscopic techniques to indicate the coordination site of the ligand **H3**, which was through the nitrogen in the C_23_=N bond of the azasterol; noticeable stability was observed in DMSO. ZnCl_2_(**H3**)_2_ was evaluated against *L. amazonensis* promastigotes and intracellular amastigotes, and promising results were acquired. Regarding the results of the biological studies, the coordination of the azasterol **H3** to Zn(II) ion increased the activity in both parasite developmental stages. Furthermore, ROS production and mitochondrial membrane potential analysis confirmed the significant ultrastructural alterations observed in the promastigotes. The structural lesions observed in the endoplasmic reticulum and mitochondrion resulted in a significant increase in ROS production and decrease in the mitochondrial membrane potential. Additionally, the analysis of free sterol in *L. amazonensis* promastigotes exhibited the percentage accumulation of cholesta-type sterols using different concentrations of **H3** and ZnCl_2_(**H3**)_2_ to inhibit the ergosterol biosynthesis. Nonetheless, ZnCl_2_(**H3**)_2_ was able to promote a 20% higher cholesterol accumulation compared to the free ligand **H3**. Thus, both compounds could inhibit 24-SMT by observing the depletion of endogenous sterols in parasite cells. This work reveals the high potential of **H3**, particularly its Zn derivative, for treating leishmaniasis, providing insights into the mechanisms of action involved.

## 5. Patents

Visbal, G., Navarro, M., Justo, R. M. S., Rodrigues, J. C. F., Silva, S. T. M., De Souza, W., Da Silva, G. S. Delta 24-sterol methyltransferase enzyme inhibitor metalloazasterol compound, pharmaceutical composition, and use of the compound. Brazilian-granted patent BR 10 2020 019121-7 A2/WO 2022/061431 A1.

## Data Availability

Supporting information is available enclosed.

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
