# Peer review of "Zinc(II)-Sterol Hydrazone Complex as a Potent Anti-Leishmania Agent: Synthesis, Characterization, and Insight into Its Mechanism of Antiparasitic Action"

_pharmaceutics, 2023, doi:10.3390/pharmaceutics15041113_

Round 1

Reviewer 1 Report

1- The Ligand is represented in the manuscript as H3 which is not applicable, the authors have to change it in all parts of the manuscript to the proper way such as LH3.

2- There is no result of the molar conductance of the ligand itself in the result even though it's supposed to be done.

Author Response

Referee 1:

1- The Ligand is represented in the manuscript as H3 which is not applicable, the authors have to change it in all parts of the manuscript to the proper way such as LH3.

Response: We appreciate the reviewer’s comments. But respectfully, we prefer to represent the ligand studied in this work as H3, consistently as it has been identified in our previous articles.

2- There is no result of the molar conductance of the ligand itself in the result even though it's supposed to be done.

Response: As requested, the molar conductance of the ligand (H3) was determined, and the value obtained was ΛM = 1.80 ohm-1cm2mol-1, as expected for a neutral compound.

Reviewer 2 Report

the manuscript entitled "Zinc(II)-sterol hydrazone complex as a potent anti-Leishmania agent: Synthesis, characterization and understanding of its antiparasitic mechanism of action." aims to provide the synthesis and characterization of a new complex ZnCl2(H3)2 or also called here ZnH3, as well as its biological activity against Leishmania amazonensis. The study was conducted with good scientific rigor and well structured. Several methods were addressed to better elucidate the anti-Leishmania activity of the compounds under study. In addition, the topic of the paper falls within the scope of the journal, so I recommend publication only after the authors resolve these observations:

Line – 48 : change leishmania sp with spp.

Remove space in line 107

Lines 150-151 include “ of infected mice” after lesions.

Lines 161-164: Why did you add the compounds 24 hours after the Leishmanias were sown and not at time 0? Also, it would be appropriate to include the concentrations of the compounds used in this section.

Lines 164-167: what culture medium did you use for the monocytes? it would be good to specify.

Lines 170-172: it is appropriate to report the concentrations of the compounds also used in this part.

Lines 381-382: write in italics L. amazonensis, furthermore, the sentence seems incomplete

Line 496: change shows with showed

Lines 634-635: Add number of subsection “3.3.” of the results.

Author Response

Referee 2:

The manuscript entitled "Zinc(II)-sterol hydrazone complex as a potent anti-Leishmania agent: Synthesis, characterization and understanding of its antiparasitic mechanism of action." aims to provide the synthesis and characterization of a new complex ZnCl2(H3)2 or also called here ZnH3, as well as its biological activity against Leishmania amazonensis. The study was conducted with good scientific rigor and well structured. Several methods were addressed to better elucidate the anti-Leishmania activity of the compounds under study. In addition, the topic of the paper falls within the scope of the journal, so I recommend publication only after the authors resolve these observations:

Response. The authors thank the reviewer for the kind words and the approval of our manuscript for publication.

Line – 48: change leishmania sp with spp.

Response: It was corrected in the revised manuscript.

Remove space in line 107

Response: It was removed in the revised manuscript.

Lines 161-164: Why did you add the compounds 24 hours after the Leishmanias were sown and not at time 0? Also, it would be appropriate to include the concentrations of the compounds used in this section.

Response: Our decision to add the drugs 24h after initiating the culture cells is related to giving time for the parasites to establish to the new culture medium before adding the drugs. We included in the revised manuscript the concentrations used for each compound, as recommended.

Lines 164-167: what culture medium did you use for the monocytes? it would be good to specify.

Response: The peritoneal lavage was carried out with Hank’s solution and the macrophage cultures were maintained with RPMI. These details of the methodology were added in the revised manuscript.

Lines 170-172: it is appropriate to report the concentrations of the compounds also used in this part.

Response: It was added in the revised manuscript.

Lines 381-382: write in italics L. amazonensis, furthermore, the sentence seems incomplete.

Response: It was written correctly. The sentence was revised and completed, numbering it as subsection.

Line 496: change shows with showed.

Response: As suggested, it was changed

Lines 634-635: Add number of subsection “3.3.” of the results.

Response: The number of this subsection is 3.3.2 in the revised manuscript.

Reviewer 3 Report

This manuscript "Zinc(II)-sterol hydrazone complex as a potent anti-Leishmania agent: Synthesis, characterization, and insight into its mechanism of antiparasitic action" presents an interesting and comprehensive study on the synthesis, characterization and biological evaluation of a new ZnCl2(H3)2 complex against Leishmania amazonensis. The work provides detailed information on the spectroscopic and analytic techniques used to characterize the complex, as well as its biological activity against the parasite. The results from the biological studies indicate that the ZnCl2(H3)2 complex was seventeen times more potent than the free ligand H3 against the intracellular amastigote stage, with a high selectivity index. Additionally, the electron microscopy studies revealed that the complex was able to induce significant membrane wrinkle, mitochondrial injury, and abnormal chromatin condensation changes. The authors also provide evidence of the complex's ability to inhibit the ergosterol biosynthesis.

Overall, this is a well-structured and well-executed work that presents a promising new alternative for treating leishmaniasis. The authors should be commended for their efforts and the manuscript can be accepted for publication with minor revisions.

Minor revisions:
1. Reword the introduction to provide a clearer overview of the study.

2. Include more discussion on the stability of the complex and how it can be used for drug delivery.

3. The authors should provide more information on the possible reasons ZnCl2(H3)2 interaction mechanism with hydrazones. In my view, this could be attributed to the prevalence of defined conformers of molecules, as evidenced in the literature for the same componds [Gamov, G.A., et.al(2019) Journal of Molecular Liquids, 283, pp. 825-833. DOI: 10.1016/j.molliq.2019.03.125 ; (2021) Physics and Chemistry of Liquids, 59 (5), pp. 666-678. DOI: 10.1080/00319104.2020.1774878 ;(2021) Journal of Molecular Liquids, 342, № 117372, . DOI: 10.1016/j.molliq.2021.117372].Please discuss this point in the text of manuscript.

4. Include more discussion on the stability of the complex and how it can be used for drug delivery for conclusions.

Author Response

Referee 3:

This manuscript "Zinc(II)-sterol hydrazone complex as a potent anti-Leishmania agent: Synthesis, characterization, and insight into its mechanism of antiparasitic action" presents an interesting and comprehensive study on the synthesis, characterization and biological evaluation of a new ZnCl2(H3)2 complex against Leishmania amazonensis. The work provides detailed information on the spectroscopic and analytic techniques used to characterize the complex, as well as its biological activity against the parasite. The results from the biological studies indicate that the ZnCl2(H3)2 complex was seventeen times more potent than the free ligand H3 against the intracellular amastigote stage, with a high selectivity index. Additionally, the electron microscopy studies revealed that the complex was able to induce significant membrane wrinkle, mitochondrial injury, and abnormal chromatin condensation changes. The authors also provide evidence of the complex's ability to inhibit the ergosterol biosynthesis.

Overall, this is a well-structured and well-executed work that presents a promising new alternative for treating leishmaniasis. The authors should be commended for their efforts and the manuscript can be accepted for publication with minor revisions.

Response: The authors thank this reviewer for the kind words, and encouragement and for appreciating the quality of our manuscript.

Minor revisions:

  1. Reword the introduction to provide a clearer overview of the study.

Response: The introduction was revised and reworded as recommended.

  1. Include more discussion on the stability of the complex and how it can be used for drug delivery.

Response: The discussion of the stability of ZnCl2(H3)2 complex was rewritten. We appreciate the comment regarding how this compound can be used for drug delivery, unfortunately, we do not have the response so far, further studies need to be done, particularly in vivo assays, which are planned to be performed.

  1. The authors should provide more information on the possible reasons ZnCl2(H3)2 interaction mechanism with hydrazones. In my view, this could be attributed to the prevalence of defined conformers of molecules, as evidenced in the literature for the same componds [Gamov, G.A., et.al(2019) Journal of Molecular Liquids, 283, pp. 825-833. DOI: 10.1016/j.molliq.2019.03.125 ; (2021) Physics and Chemistry of Liquids, 59 (5), pp. 666-678. DOI: 10.1080/00319104.2020.1774878 ;(2021) Journal of Molecular Liquids, 342, № 117372. DOI: 10.1016/j.molliq.2021.117372]. Please discuss this point in the text of manuscript.

Response: The recommended articles show important studies such as a) thermodynamics and kinetics formation of hydrazone derivatives from pyridoxal 5′-phosphate, and hydrazides 2-furoic, thiophene-2-carboxylic hydrazides in solution; b) conformations studies of pyridoxal 5ʹ-phosphate derived Schiff bases by NMR and quantum chemical calculations; c) complexation of these hydrazones with three cations (Co2+, Ni2+, Zn2+); and d) its stability in aqueous solutions studies. All these studies were performed in order to get hydrazones derived from pyridoxal 5′-phosphate as a fluorescent indicator and the potential biological application as the agent of heavy metal detoxification.

The referenced articles were not deemed to be related to the topic of the current manuscript, which evaluates the antileishmanial potential of ZnCl2(H3)2 complex using a synergism strategy. Knowing that 22-hydrazone- imidazolin-2-yl-chol-5-ene-3β-ol (H3) is an inhibitor of 24-SMT (references 15 and 16 in the revised manuscript), an enzyme that plays an essential role in parasite sterol biosynthesis, it was coordinated to Zn(II) in order to enhance its activity and decrease its toxicity.

On the other hand, we agree that hydrazones can adopt different conformers. This conformational study has been reported by us using several azasterols, among them H3. In fact, this study suggested the importance of the conformational flexibility introduced by the extra methylene group and the stereo-electronic distribution in the side chain of the sterols in 24-SMT inhibition, leading to a high antiproliferative effect by disrupting ergosterol homeostasis.

4. Include more discussion on the stability of the complex and how it can be used for drug delivery for conclusions.

Response: The stability discussion was added in the conclusion, as recommended.